# Morphological Diversity and Dynamics of Dengue Virus Affecting Antigenicity

**DOI:** 10.3390/v13081446

**Published:** 2021-07-24

**Authors:** Guntur Fibriansah, Xin-Ni Lim, Shee-Mei Lok

**Affiliations:** 1Programme in Emerging Infectious Diseases, Duke–National University of Singapore Medical School, Singapore 169857, Singapore; gfibriansah@duke-nus.edu.sg (G.F.); xinni.lim@duke-nus.edu.sg (X.-N.L.); 2Centre for BioImaging Sciences, Department of Biological Sciences, National University of Singapore, Singapore 117557, Singapore

**Keywords:** dengue, flavivirus, morphological changes, antibody complex

## Abstract

The four serotypes of the mature dengue virus can display different morphologies, including the compact spherical, the bumpy spherical and the non-spherical clubshape morphologies. In addition, the maturation process of dengue virus is inefficient and therefore some partially immature dengue virus particles have been observed and they are infectious. All these viral particles have different antigenicity profiles and thus may affect the type of the elicited antibodies during an immune response. Understanding the molecular determinants and environmental conditions (e.g., temperature) in inducing morphological changes in the virus and how potent antibodies interact with these particles is important for designing effective therapeutics or vaccines. Several techniques, including cryoEM, site-directed mutagenesis, hydrogen-deuterium exchange mass spectrometry, time-resolve fluorescence resonance energy transfer, and molecular dynamic simulation, have been performed to investigate the structural changes. This review describes all known morphological variants of DENV discovered thus far, their surface protein dynamics and the key residues or interactions that play important roles in the structural changes.

## 1. Introduction

Dengue virus (DENV) is one of the most important viruses transmitted to humans by the bite of the infected Aedes mosquitoes [1]. DENV is a member of the flaviviridae family, along with the other major human pathogens such as Zika, West Nile, Japanese encephalitis, and yellow fever viruses. DENV can cause a range of illnesses with different severities, from asymptomatic, to mild dengue fever, to severe dengue hemorrhagic fever (DHF) or dengue shock syndrome (DSS). It is estimated that 284–528 million DENV infections occur annually, resulting in 67–136 million cases of dengue fever with symptoms [2]. A live attenuated vaccine, the chimeric yellow fever 17D-tetravalent dengue vaccine (CYD-TDV), has been licensed by several dengue-endemic countries in Asia and Latin America. However, its use is limited to the population above 9 years old, with poor efficacy against DENV serotype 2 [3,4]. There are no approved antiviral drugs to treat DENV infection [5,6].

DENV consists of four different serotypes (DENV1-4) [7,8]. The four DENV serotypes differ by ~25–40% in the amino acid sequence of the polyprotein [9]. Within a serotype, DENV strains have up to 3% amino acid variation and can further be grouped into genotypes [10,11]. Secondary infection by a heterologous serotype has been associated with a severe disease, i.e., DHF or DSS. This severe disease is associated with the formation of the complex of non-neutralizing or sub-neutralization concentration of antibodies (elicited from the previous infection), with DENV leading to enhanced viral infection through a mechanism known as antibody-dependent enhancement (ADE) [12]. This makes the development of both antibody therapeutics and vaccines difficult. Further complicating this, our lab and others showed that the strains within a serotype can have different morphologies, hence affecting its antigenicity properties.

## 2. DENV Particle Composition and Its Infection Cycle

The inside of the mature DENV particle contains a positive-sense single stranded 10.7 kb genomic RNA complexed with capsid (C) proteins [13]. This is surrounded by a bilayer lipid membrane anchored with membrane (M) and envelope (E) proteins [13,14]. The E protein is the major target for neutralizing antibodies [15]. It consists of three domains: DI, DII and DIII [16,17,18,19]. A hydrophobic fusion loop, which is located at the distal end of DII, plays an important role in membrane fusion between the viral lipid bilayer and the host endosomal membrane during virus entry into the cell [20,21]. DIII and glycosylation at residues N67 are known to be the interaction sites for receptors [22].

The infection cycle of mature DENV into cells [23] starts with the attachment of mature virus to cell receptor such as DC-SIGN [24], TIM1 [25], etc. The virus is then endocytosed inside the cell [26], and the low pH of the endosome causes fusion of the viral membrane with the endosomal membrane [20,27]. This releases the viral genome into the cell cytoplasm allowing viral genome replication. New immature DENV (immDENV) particles are assembled in the endoplasmic reticulum (ER), and then the virus undergoes maturation when moving through the trans-golgi network [28,29]. On the surface of the immDENV, in addition to the E proteins, there is the precursor-membrane protein (prM) [14,30]. The maturation process involves the immDENV, dramatically rearranging its surface quaternary structure from a spikey to a smooth-surfaced particle, and the removal of the pr portion of the prM, thus producing the fully mature virus with E and M proteins [31].

### Maturation Process of DENV–How Immature DENV Is Processed into Mature DENV Particles

ImmDENV is assembled in the neutral pH environment of the ER. The C, prM and E proteins are the first proteins being translated by ribosomes on the cytoplasmic side of the ER membrane [32,33]. The newly translated C, prM and E proteins are then threaded back and forth through the ER membrane via their transmembrane (TM) regions. The final location of the C proteins are in the cytoplasmic side of the ER, while the E and M proteins have their transmembrane region anchored on the ER membrane facing the lumen side of the ER [34]. The E and M proteins form heterodimers, and three of these heterodimers will subsequently form trimeric inverted tripod-like structures (Figure 1a,b) [35]. These inverted tripods criss-cross and interact with each other, forming a network (Figure 1b) that will later become the surface of the immDENV [14,30]. On the cytoplasmic side of the ER, C proteins were translated into immature C protein, which contains five helices (α1 to α5). The helix α5 is then cleaved off by viral NS2B-NS3, forming the mature C protein [36]. However, we have shown previously that there is a presence of a mixture of mature and immature C proteins inside immature particles from a closely related flavivirus, Zika virus [35]. Both immature and mature C proteins can assemble into dimers (Figure 1b) [34,35]. The helix α5 of the immature C protein helps the nucleation of the assembly of the C protein dimers to form triangular networks [34,35]. These C protein assemblies interact with the TM regions of the prM-E proteins, thus modulating the quaternary arrangement of the outer surface proteins (prM-E). They form a bridge between the genomic RNA and the TM regions of prM-E proteins [34,35], ensuring that all immature virus particles contain viral RNA genome. Once the virus particle has assembled its RNA genome, the particle then buds off the ER membrane.

The newly formed immDENV contains 180 copies of prM-E proteins assembled into 60 copies of inverted tripod structures, thus making the viral surface look spiky (Figure 1a) [14,30]. The immDENV surface protein arrangement follows the icosahedral symmetry with one asymmetric unit consisting of three prM-E proteins [molecules (mols) A, B and C]. These immDENV particles are then transported through the trans-golgi network where the pH environment gradually becomes acidic [37]. The acidic pH triggers massive viral surface rearrangement, changing the morphology of the virus from a spiky to a smooth surface (Figure 1a) [31]. This structural arrangement involves the trimeric prM-E proteins to disassociate from each other and then reassociating to form homodimers (Figure 1c) [31]. The resultant structure also causes the furin-cleavage site located on the stem between the pr molecule and the M protein to become more exposed, allowing cleavage to occur [31]. However, the cleaved pr molecule remains attached to the E protein due to the low pH environment of the exosomes (Figure 1a) [31,37,38,39]. This ensures that the partially processed immature virus will not fuse back into the cell. It is unknown if the pr molecule help to stabilize the smooth virus surface structure; however, superposition of the prM+E protein complex onto the high resolution spherical compact mature virus particle suggests that the pr will interfere with the interaction of the E protein fusion loop with the DI-DIII domain of the opposite E protein protomer within a dimer, suggesting that the presence of pr does not stabilize the smooth virus structure. Indeed, Yu et al. [31], shows that the switching of pH conditions between neutral and low pH, and then back to neutral pH, will switch the conformation of the uncleaved immature virus with prM from spikey to smooth and then back to spikey surface morphology, suggesting that the presence of pr does not stabilize the smooth partially processed immature virus surface. Therefore, there must be another factor that helps to maintain the smooth surface structure of these immature virus at acidic pH. Zhang et al. [40] showed that the interaction of the membrane-associated M part of the prM protein with the E protein is enhanced at low pH compared to neutral pH—this may help to “attract” the E proteins to lay flat on the virus surface, causing the smooth surface appearance. The last step of the maturation process involves the release of the pr molecule when the virus is exported into the neutral pH extracellular environment; the increase in pH reduces the affinity between the pr and the fusion loop of the E protein, thus forming the fusion-competent mature virus particles [31].

## 3. The Different Morphological Variants and Their Dynamics

We often observed a mixture of mature, partially mature and highly immature virus particles in a mature virus preparation [41,42]. This suggests that the maturation process is inefficient, but what causes this is largely unknown. The important factors in the maturation process are: the acidic pH conditions in the trans-golgi network that stimulate the rearrangement of surface proteins on immature dengue virus, and also the furin protease activity required for cleavage of pr from the prM molecule. It is possible that these are conditions are not optimal leading to the partial processing of immature dengue virus particles.

As even the highly immDENV has been shown to be infectious to DC-SIGN-bearing cells [43] and also to Fc-gamma receptor positive cells when immDENV is complexed with some (weakly or non-neutralizing antibodies) anti-E and anti-prM antibodies [44,45], therefore all of viruses with different degree of maturation are considered infectious. In addition, we also observed that the fully mature DENV strains can experience a change in morphology (expanded spherical [46] and clubshape morphology [47]) after incubation at higher temperature. It is hypothesized that this will further introduce morphological diversity for the virus to evade immune response. The following sections discuss the morphological diversity, their dynamics and how they are infectious and could evade immune response.

### 3.1. ImmDENV

In the immDENV spiky neutral pH structure, the trimeric prM:E spikes are criss-crossing on the virus surface. Parts of the lipid bilayer are exposed, particularly on the 5- and 3-fold vertices (Figure 1a). The prM-E proteins are tilted ~45 ° on the surface of the immDENV and they interact at the top of the spike where the pr molecules are assembled. Overall, the immDENV surfaces are structurally more labile and thus are determined as being low resolution-DENV1 (6 Å resolution, PDB ID 4B03) [14], DENV2 (12 Å, 1TGE) [19], WNV (24 Å, 2OF6) [48], ZIKV (9 Å, 6LNU) [35], Spondweni virus (7.8 Å, 6ZQW) and Binjari virus (4.4 Å, 7L30) [49].

It is speculated that the spiky immDENV, after gaining entry into the cell, will undergo massive structural surface rearrangement in the endosome, stimulated by the low pH environment—this motion will likely be similar to that of the newly synthesized immDENV in the cell when it undergoes maturation in the acidic transgolgi-network. The endosome also contains furin protease, and thus will allow cleavage of the prM. However, in the last step of the normal immature virus maturation process, an increase in pH is required (when the partially processed immature virus is released to the neutral pH extracellular environment) for the pr to detach from the E protein fusion loop, but this is not possible when the virus is in the endosomes, as the environment will continue to become more acidic with time. The question is then how the immDENV when complexed with anti-prM antibody could help to overcome this in the endosome. Wirawan et al. [45] showed that when the Fab fragment of an anti-prM is bound to highly immature DENV, the Fab introduced extra steric hindrance to the already massive structural changes of the surface proteins that are induced at low pH; this helps to “knock off” the prM molecule that is capping the virus E protein fusion loop. This may thus help the immDENV to fuse with the endosome [45].

### 3.2. Mature DENV Particles

Some DENV serotypes and strains generate high percentage of mature compact DENV particles, e.g., DENV2 strain PVP94/07 [50] and DENV4 strain 06K2270DK1 [51]. Other strains produce mature virus with the compact smooth surface DENV at mosquito physiological temperature (28 °C), but this changes to a bumpy surface structure when incubated at a higher temperature (37 or 40 °C) [46,50]. One example is DENV2 strain NGC, that has been passaged in C6/36 mosquito cell line extensively. However, when this virus strain is passaged in BHK cells, the particles seem to have a smooth surface morphology at 37 °C. Other strains can also change to a non-spherical clubshape particle morphology [47].

#### 3.2.1. Mature Compact DENV Structure

On the surface of the mature compact structure, there are 180 copies of E proteins arranged as 90 head-to-tail homodimers (Figure 1a), and three of the homodimers are aligned parallel to each other, forming a raft structure [13,52]. These E proteins are tightly packed on the virus surface (Figure 1a). Most of their periphery residues are involved in making contact with adjacent E proteins either to form a dimer (intradimer), interdimer or inter-raft interactions, leaving no gaps on the virus surface. Due to this compactness, mature virus structures have been solved to a higher resolution compared to the immature virus structures—DENV1 (4.5 Å, 4CCT) [14], DENV2 (3.1 Å, 6ZQU), DENV3 (6 Å, 3J6S) [53], DENV4 (4.1 Å, 4CBF) [51], ZIKV (3.7 Å, 5IZ7) [54], WNV (10 Å, 3J0B [55]/3.1 Å, 7KVA [56]) and chimeric DENV2/Binjari virus (2.5 Å, 7KV8 [56]).

#### 3.2.2. DENV Bumpy Surface Structure at 37 °C

DENV can change from smooth to bumpy-surfaced particles when incubated at 37 or 40 °C, as observed in DENV1 strain WestPac74 and DENV2 strains NGC (Figure 2a) [46], S16803 [46] and S16681 [57] (Table 1). Using DENV2 strain NGC particles at 37 °C [46], cryoEM reconstruction shows four structural classes of particles with Class I similar to that of the normal smooth surface compact mature DENV particle, whereas Class II to IV are particles with their surface E proteins gradually loosening their interactions with each other resulting in particles with various bigger diameters. Class I forms ~16% of the virus population. The cryoEM map shows that the particle has a radius of ~240 Å and a relatively smooth surface with small bumps at locations corresponding to the DIII of E proteins. The central cross-section of the cryoEM map showed the E protein ectodomains remained in contact with the trans-membrane (TM) helices of E and M proteins. The TM regions of both proteins embedded in the lipid bilayer were resolved, indicating limited movement on this region. Class II particles, which form ~34% of the population, are rough looking particles with protrusions located between the 5- and 3-fold vertices. These particles are slightly larger, with a radius of ~250 Å. Class III particles, which form ~39% of the population, have the largest radius, of ~260 Å. There are protrusions located between the 5- and 3- fold vertices, and these protrusions are positioned at a radius of 280 Å from the center of the particle. Additionally, there is a hole at all of the 3-fold vertices, thus exposing the lipid bilayer membrane underneath the E protein layer. Class IV particles, which form ~10% of population, have the smallest radius of ~180 Å, because most of the E protein cryoEM densities are missing. This suggests that the E proteins become very flexible, and the surface E proteins are no longer organized in icosahedral symmetry and hence are averaged out during the image reconstruction procedure.

The gaps between the E protein layer and the lipid bilayer of Class II and III particles are likely caused by the E protein ectodomains loosening their interactions with neighboring E proteins and also the E and M protein TM helices on the viral lipid membrane. The Class III cryoEM map (Figure 2b) [46] shows that all E proteins moved outwards and were rearranged. The E protein mols A and C′, located near the 5-fold vertices, remained as a dimer, whereas the protomers of the B-B′ dimer, at the 2-fold vertices, dissociated from each other.

#### 3.2.3. Determinant(s) on E Protein That Causes Bumpy Surface Structure

The DENV2 strain NGC, which changed from a smooth compact surface to a bumpy surface morphology when the temperature is switched from 28 to 37 °C, has a long passage history in mosquito C6/36 cell lines grown at 28 °C. Meanwhile, the surface morphology of the same NGC strain that was passaged in mammalian BHK21 cells at 37 °C (Table 1) [58] remained smooth and compact. Comparison of the smooth-surfaced NGC with the bumpy-surfaced NGC at 37 °C showed substitutions of five amino acid residues: I6M, D71E, G112S, N124I, and I402F [58]. Residues 6, 71 and 402 are highly/relatively conserved changes. In the compact mature DENV2 structure (PDB ID: 3J27) [52], residues 71, 112, 124 and 402 are not involved in any E–E interactions (Figure 2c). Only residue 6 is involved in the E protein intradimer interaction, interacting with residues on fusion and also ij-loops of the opposite E protein protomer within a dimer. A mutational study was completed by using an infectious DENV2 strain NGC clone with a smooth surface morphology to determine which residues when mutated can cause virus to change to bumpy surface at 37 °C. A mutant which contains five substitutions on the E protein (I6M, D71E, G112S, N124I and I402F) becomes bumpy at 37 °C [58]. Other mutants, including (1) a mutant with E protein mutations at I6M, D71E, G112S and I402F and (2) a mutant with D71E, G112S and I402F, also turned bumpy at 37 °C. The mutant with a single substitution (I6M) at the E-E intra-dimer interface also leads to bumpy-surfaced particles at 37 °C [58]. Thus, the results show that there are redundancies in amino acid residues that can result in smooth to bumpy surface morphology change.

Examination of four naturally circulated DENV2 strains (05K4155, PVP94/07, PVP103/07 and SL56) showed that at 37 °C, the surface of three of these DENV2 strains (05K4155, PVP94/07 and SL56) remained smooth, whereas strain PVP103/07 turned into the bumpy morphology [58]. At 40 °C (mimicking high fever), all clinical strains turned bumpy. DENV2 strains PVP94/07 and PVP103/07 were isolated in the same year (2007) from the same outbreak. They differ only by one amino acid at position 262 (threonine vs. methionine), yet the morphology at 37 °C is different. The T262M substitution is located at the E protein intra-dimer interface (Figure 2d), as does the I6M mutation identified in the DENV2 strain NGC mutant, which also changed the virus morphology from smooth- to bumpy-surfaced [58]. These findings suggest that destabilization of the dimeric interaction between E protein protomers is important in conferring the bumpy surface morphology. Molecular dynamics (MD) simulations with these substitutions have been performed to examine the dimeric interaction between the E protein protomers [58]. Results show that residue 6, which is located on a loop in DI, changes the secondary structure of the loop in mutated I6M, causing a cascade of global changes across the E protein dimer structure [58]. DI was observed to shift followed by a reorientation of DII. One of the protomer within the dimer appears to undergo conformational changes earlier than the other. This asymmetric behavior suggests that the movement would disrupt the symmetry-related contacts on the smooth virus surface, resulting in bumpiness at 37 °C [46]. Similar E protein structural destabilization was also observed in the MD simulation for T262M substitution. The mutation diminishes the hydrogen bond interaction between residues 262 and 258 within a E protein protomer, leading to the destabilization of the helical structure located at the middle of the intra-dimeric interaction between the E protomers, thus disrupting the symmetry-related contacts.

Most lab-adapted viruses, with a long passage history in C6/36 mosquito cell lines at 28 °C, have a bumpy surface morphology, while the clinical DENV isolates which circulate between human and mosquito hosts are largely smooth surface particles [46,58]. The observation that many different E protein mutations [58] can result in bumpy surface particles at 37 °C suggests that during the passaging of a virus numerous times in mosquito cell lines at 28 °C, the virus accumulates some mutations that may lead to it acquiring the bumpy surface morphology. In the laboratory conditions where there is little selection pressure, these bumpy surface particles outgrow the smooth virus surface particles. In contrast, it was observed that three out of four clinical DENV isolates are smooth-surfaced, suggesting there is selection pressure through circulating between the mosquito and human hosts, which favors the smooth surface particles [58]. This also indicates that in the natural environment, the bumpy particle is perhaps not the most optimal for dengue transmission. However, one clinical strain has become bumpy-surfaced, suggesting that although these particles are likely not optimal, in certain special circumstances, e.g., to evade existing immunity, the change to bumpy morphology is necessary for survival. Indeed, it was observed that DENV2 strains PVP94/07 and PVP103/07, which were from the same outbreak, have an almost identical E protein sequence, but PVP103/07 virus has acquired one mutation that can turn it into bumpy-surface particles [58].

## 4. Dynamic Motions of DENV Particles

The low resolution cryoEM maps of bumpy surfaced DENV2 strain NGC at 37 °C indicate that the E proteins on virus surface are moving, since the E protein densities are weak and “smeared”. This does not allow us to observe the dynamic motions of structural proteins of the virus. Hence, the amide hydrogen-deuterium exchange mass-spectrometry (HDXMS) method was used to identify important regions in E, M or C proteins at peptide resolution that could play important roles in the structural changes [60]. This technique involves incubating the virus in deuterated water. When regions of protein moved, their solvent accessibility increased, the deuterium in the water then replaced the hydrogen atoms on the backbone of those regions of the protein, leading to increased molecular weights (MW). The protein was then digested into peptides and then the profiles of all peptides from the protein were analyzed by mass spectrometry. The peptides with increased MW showed the regions of the protein that have increased motions.

The DENV surface E proteins are arranged in an icosahedral symmetry with 60 asymmetric units [13,14]. Each asymmetric unit has three individual E proteins—each of them are located near to one of these vertices, 2-, 3- or 5-fold vertices, and their local chemical environments are different [13,14]. In cryoEM determined structures, these three E proteins can be observed and differentiated clearly; however, in HDXMS, only averaged deuterium changes of all E proteins can be detected, and hence one cannot distinguish the difference in behavior between the three individual E proteins [60]. The DENV2 strain NGC, which is observed to have a bumpy surface morphology under cryoEM, seems to have more motions detected by HDXMS in E and M proteins than that of DENV1 strain PVP159 at both 28 and 37 °C [60]. The movements of the E proteins in DENV2 strain NGC are concentrated at the E-E intradimer and interdimer and also the E–M interacting interfaces consistent with the loosening of the E protein interactions with each other and the E protein shell moving to a higher radius, as observed by cryoEM [60].

In contrast with the DENV2 strain NGC, there was no differences observed in the deuterium exchange across peptides of E and M proteins in DENV1 strain PVP159 at 37 °C compared to 28 °C [60]. This supports the previous observation regarding the cryoEM images of DENV1 strain PVP159 that showed that the virus did not undergo expansion and remained smooth-surfaced at 37 °C [14,50]. The DENV1 strain PVP159 structural proteins (E, M and C) showed motions only at 40 °C [60]. On the E proteins, DI and DIII showed the highest dynamics [60], while DII at the intradimer interface showed moderate motion. Hence, the patterns of E protein dynamics at higher temperatures are very different between DENV2 strain NGC and DENV1 strain PVP159.

The movement of the whole E protein shell away from the viral lipid membrane on DENV2 strain NGC and DENV1 strain PVP159 can also be measured by using time-resolved fluorescence resonance energy transfer (TR-FRET) experiments [60]. In the TR-FRET experiment, the E proteins are labelled with AF488-TFP (a donor) and the lipid bilayer with DiI-C18 (an acceptor), and the distance between the E protein shell and the lipid membrane is monitored by the donor fluorescence lifetime. The results show that the E protein shell of DENV2 strain NGC, which has a bumpy surface morphology, gradually moves away from the viral lipid membrane when the temperature increases from 25 to 37 °C, and there are no further movements when the temperature is raised from 37 to 40 °C. These findings are consistent with the HDXMS measurements and cryoEM structures where the irreversible structural changes to bumpy surface particles at 37 °C are detected [46,60]. On the other hand, for DENV1 strain PVP159, there is no detected movement of the E protein shell. Only when incubation temperature is increased from 37 to 40 °C, then E protein shell movement is detected, consistent with the HDXMS experiments [60] and the previous cryoEM studies [50].

Combining the results from cryoEM, HDXMS and TR-FRET experiments shows that different virus serotypes and strains undergo conformational changes at different temperatures, and also these viruses moved in a different way involving different regions of the E protein on the virus surface. The intradimer E-to-E interaction at the icosahedral 2-fold vertices is shown to be the most mobile part of the DENV2 strain NGC structure, whereas for DENV1 strain PVP159, it is the interactions between E proteins around the 3-fold and/or 5-fold. This is consistent with the previous cryoEM structural analysis comparing the contacts between E-proteins in the DENV1, DENV2 and DENV4 structures [51]. It was suggested that DENV1 has stronger intra-dimeric but weaker interdimeric and inter-raft contacts compared to DENV2 and DENV4 [51]. Furthermore, DENV2 had significantly fewer charged residues at the E–M interface compared to that in DENV1 or DENV4, and thereby E protein expansion can happen easier on DENV2 [51].

## 5. Flavivirus Clubshape Particles

A new morphological flavivirus variant adopting a temperature-induced clubshape (ClubSP) morphology has been discovered [47]. It usually exists as a minor virus population in flaviviruses (DENV1, DENV2 and ZIKV), except for DENV3 strain CH53489, for which almost half of its virus population has the ClubSP morphology (Figure 3a) [47]. The ability of even some flavivirus particles to change to clubshape morphology shows they could possibly morph into this dramatically different structure if there is a selection pressure, e.g., to escape from antibody binding. Detailed characterization of the ClubSP has been carried out on the DENV3 strain CH53489. It has been observed that the ClubSP forms from the round spherical shape particles within 15 min of incubation at 37 °C [47]. The DENV3 strain CH53489 grown in both mosquito and mammalian cells has the ability to form ClubSP [47]. It has been demonstrated that the original RNA genome-filled smooth particles, after becoming ClubSP at 37 °C, did not lose their RNA genome [47]. The clubSP also has the ability to attach onto mosquito C6/36 or mammalian BHK cells. Together, this evidence suggests that clubSP is likely infectious.

The clubSP contains a head and a tail. The sizes of the head and the length of the tail vary between particles [47]. However, the tails have the similar diameter, suggesting that there could be some structural order, e.g., helical symmetry, that could allow the alignment and averaging of the central segments of the tail for structural determination. The 2D classification of the central segments of the tails showed they are smooth-surfaced with a diameter of ~200 Å and the Fourier transform of the 2D average suggests there is no helical symmetry [47]. This could indicate that the structure could be either be disordered or without helical symmetry, or the segments were just simply misaligned to each other due to a lack of obvious features for alignment. Fab C10 was used to complex with DENV3 strain CH53489 to help stabilize the clubSP structure and also provide additional features for more accurate alignment (Figure 3b). The tail of the Fab C10:clubSP complex indeed showed helical symmetry. The uncomplex Zika virus, when heated to 37 °C, induced some clubSP [47], but when Zika virus is first complexed with Fab C10 and then incubated at 37 °C, those particles instead adopt a “caterpillar” shape, hereafter named as catSP. The helical structures of DENV3 CH53489:C10 clubSP and ZIKV:C10 catSP were reconstructed to 9.4 and 10.4 Å resolution, respectively [47]. ClubSP and catSP have different helical parameters consistent with the differences in the diameters of the tail of ClubSP and the body of catSP. The central segments of the helical tail of DENV3 CH53489:Fab C10 clubSP have outer and inner diameters of 150 and 30 Å, respectively (Figure 3b). The Fab ZIKV-Fab C10 catSP has an outer diameter of 270 Å (excluding the Fab molecules) and an inner diameter of 130 Å (Figure 3c). For both structures, their helical asymmetric unit consists of two Fab C10 molecules bound to an E protein dimer (Figure 3b,c). The E protein dimers of the tail of DENV3 CH53489-Fab C10 clubSP are arranged differently compared to the ZIKV-Fab C10 catSP structure. In the DENV3 CH53489-Fab C10 clubSP tail structure, the Fab molecules bind across the protomers within a E protein dimer, thus locking them together. However, the inter-dimer interactions are free to move, and hence the dimers have shifted away from each other compared to the original E protein raft structure in the spherical smooth surface particles. This also indicates that Fab C10 does not bind across the E protein inter-dimer interface on DENV3. The E protein ectodomains moved outwards, away from the viral lipid membrane, indicating that their interactions with their own stem and transmembrane region are loosened. This is consistent with the curvature of the E protein dimer in the clubSP structure that more resembles the flatter curvature of the E protein ectodomain crystal structure complexed with Fab C10 [47] than the E proteins on the surface of the smooth compact spherical virus structure. Incubation of spherical DENV3 strain CH53489 to higher temperatures of 29 °C or 37 °C would provide sufficient energy to loosen both the E protein inter-dimers and inter-raft interactions. The raft structures undergo some rotations with concomitant translation of E protein dimers along their main axis relative to the other dimers in the same raft, completing the formation of the helical structure (Figure 3b).

The helical structure of the Fab ZIKV-Fab C10 catSP shows the E protein dimers arranged parallel to each other in an identical way to that observed between the E protein dimers within a raft of the ZIKV-Fab C10 spherical virus complex structure. In the 4 Å resolution spherical ZIKV-Fab C10 complex structure, Fab C10 not only binds across E protein protomers at the intradimer interface, but it can also lock different E protein dimers together by engaging residues across the inter-dimer interface [61]. Thus, the E protein raft containing three dimers is locked together as a rigid body and only relative motions between rafts are possible. Thus, this could explain the morphological shift of the Fab C10:ZIKV catSP structure, where the E proteins within the raft are locked together and then the rafts rotate relative to each other to form the same type of interactions as observed between dimers at the inter-dimer interface (Figure 3c).

## 6. DENV4 Structure Displays the Most Stable Structure at Temperature of 40 °C

Among all DENV serotypes, DENV4 strain 06K2270DK1 has been reported to be the most stable virus structure so far (Table 1) [51]. Similar to DENV1 strain PVP159, DENV4 strain 06K2270DK1 did not show any morphological changes at 37 °C under cryoEM. Interestingly, after 30 min incubation at 40 °C, the surface of the virus particles mostly remained smooth; however, there was slight aggregation of the particles. Compared to DENV2 strain NGC, DENV4 strain 06K2270DK1 has significantly more electrostatic interactions and potential hydrogen bonds on the E–M interface [51]. Due this stronger E–M interaction, the E protein ectodomain does not dissociate and move to a higher radius at a higher temperature, different to what has been observed on DENV2 strain NGC [46,58].

## 7. Virus Morphological Diversity Influences Antibody Binding

The ability to change into different morphologies and thus their antigenicity properties suggest the virus could use this as a way to escape from the immune system [47]. Below is a discussion of how antibodies bind to different types of morphologies and their influence on the neutralization activity.

Previously, it has been shown that serotype-specific highly neutralizing human monoclonal antibodies (HMAb), such as 1F4, 5J7 or 14c10, recognize the quaternary-dependent epitope presented only on the smooth compact DENV surface [53,62,63]. Their binding epitopes are either across two or more E proteins or require specific curvature of the E protein that are presented on the surface of the smooth compact DENV surface. This suggests that such antibodies may not be able to bind “bumpy” surface mature DENV, the clubshape particles, or the partially and fully immature virus well, as their quaternary dependent epitope will be disrupted in these morphologies.

The ClubSP particles have a head and a tail, and hence the E proteins might have different antigenic characteristic on different parts of the particle. Several antibodies were tested on DENV3 strain CH53489 to determine their ability to bind and neutralize. A mouse monoclonal antibody 8A1 [64], which recognizes an epitope on DIII of E protein, was able to neutralize DENV3 strain CH53489. The Fab 8A1 has been demonstrated to be able to inhibit the formation of clubSP when the Fab was added before the virus was exposed to 37 °C [47]. When it is added after clubSP is formed, Fab 8A1 was observed to bind only to the head and the tip (but not to the body) of the tail. This suggests that the epitope on DIII is more exposed in these two regions.

Antibody 4G2, which binds to the DII fusion loop, when mixed with the clubSP particles, showed that the Fab fragment only binds to the head of the clubSP [47]. This is consistent with the inability of 4G2 to neutralize the virus even at high antibody concentrations. This result once again suggests that the ability of virus to change its morphology could help it to escape from antibody neutralization, because the antibody is unable to coat the entire virus surface.

A group of antibodies binding across E protomers within a dimer are named E-protein dimer epitope (EDE) antibodies; some of these antibodies include HMAb 2D22, C8 and C10. Their epitopes in general involve part of DIII on one E protein protomer and also the fusion loop of DII on the opposite protomer within a dimer. While HMAb 2D22 is DENV2 specific [59,65], HMAbs C8 and C10 are serotype cross-reactive, as a large proportion of their footprint is on the fusion loop, which is highly conserved across flaviviruses [66,67]. They also seemed to have the capability to bind to more diverse DENV morphologies. For example, HMAb 2D22 are able to bind to not only the smooth compact mature virus but also to the bumpy surface mature virus [59]. Its occupancy on the bumpy surface DENV is, however, lower, as the E protomers at the icosahedral 2-fold vertices are separated and are no longer dimers. Nevertheless, its potency is not compromised even with lower occupancies. HMAbs C8 and C10 are able to bind to the mature DENV and also partially immature virus [66]. These antibodies were observed to be able to bind to E ectodomain monomers in solution and then assemble them into E protein dimers [67]. Their ability to bind to partially immature viruses and neutralize them suggests that the E trimeric structures in the immature part of the viruses are assembled into dimeric forms by the antibodies. They are, however, unable to bind highly immature DENV that has a high percentage of prM molecules which will discourage dimeric interactions between E proteins. HMAb C10 was also observed to be able to bind to a large proportion of the surface of DENV3 strain CH53489 ClubSP and have shown to be very potent in neutralizing them [47]. Hence, these EDE antibodies could potentially be good therapeutic candidates against different DENV morphologies.

There are antibodies that recognize hidden or partially hidden epitopes on the compact smooth surfaced DENV. Thus, whether the antibodies could neutralize them may partly depend on the mobility/transient exposure of these epitopes. As shown by a combination of cryoEM and HDXMS, different virus strains could have different level of E protein mobility at 37 °C — some become bumpy surfaced particles with increased solvent accessibility, while others, that may appear smooth-surfaced under cryoEM, could have local mobility/dynamics of the E protein. The regions with increased dynamics on the virus surface also vary among different DENV serotypes or strains. The increased incubation temperature (normal physiological (37 °C) and high fever temperature (40 °C)) also caused increased dynamics. Indeed, at 40 °C, most DENV strains, even those exhibiting smooth compact-surfaced morphology at 37 °C, can become bumpy-surfaced, suggesting antibodies that bind well to bumpy surface particles may be a better therapeutic agent. However, for prophylactic treatment, where the antibody is injected prior to infection and hence the antibodies will likely encounter virus at 37 °C, a mixture of antibodies that bind well to both smooth compact-surfaced and bumpy-surfaced particles may work better.

Below are examples of antibodies that bind better when temperature is increased to cause viruses to either become bumpy-surfaced or to increase the E protein dynamics on smooth-surfaced virus particles. A study on West Nile virus (WNV), another flavivirus, showed that a fusion loop binding antibody, E53, has a time- and temperature-dependent neutralization profile [68]. This finding suggests that the dynamic nature of E proteins is required for binding, despite its compact appearance of the E proteins on the virus surface under cryoEM [56,68]. Another cross-reactive neutralizing mouse antibody (MAb), 1A1D-2, recognizes an epitope on DIII with 18% of the epitope hidden on the smooth compact DENV2 surface [69]. Incubating DENV2 at 37 °C for 30 min turned the virus particle into bumpy surfaced, thus increasing the solvent accessibility of E proteins, allowing the binding of the Fab 1A1D-2 [69]. The Fab binding to E protein further induced a cascade of conformational changes, allowing more Fab molecules to bind.

Another study also found that in addition to increased dynamics of E proteins on virus surface, high affinity of antibodies could help them to bind to cryptic epitopes. HMAb 1C19 binds to a partially hidden epitope (hi- and ij-loops) on DII [50]. HMAb 1C19 has been shown to have high affinity to DENV1 recombinant E protein and a much lower affinity to that of DENV2, as determined by biolayer interferometry (BLI). The antibody was then added to different strains of DENV1 and DENV2 particles that will either stay smooth-surfaced or become bumpy-surfaced at 37 °C. The cryoEM image shows that when temperature is increased to 37 °C, HMAb 1C19 was able to bind to all DENV1 strains, regardless of whether the corresponding uncomplex control becomes bumpy or not. To DENV2, since the HMAb 1C19 has a low affinity, antibody binding only occurs to the virus strain that becomes bumpy at 37 °C, and not to the one that stays smooth. The ability of HMAb 1C19 to bind to these DENV1 and DENV2 particles at different temperatures is highly correlated with their ability to neutralize. This study suggests that the inclusion of high-affinity antibodies for prophylactic and therapeutic treatment could overcome the problems associated with morphological diversity of DENV particles. Further supporting this is the observation that the poorly neutralizing antibodies that bind to cryptic epitope-fusion loop or bc loop, that formed the majority of antibodies in a primary dengue antibody response [65], can become potently neutralizing after affinity maturation in the secondary infection by another serotype [70].

## 8. Conclusions

Mature DENV particle can adopt several different morphologies. At 28 °C, most DENV particles have a spherical smooth surface morphology. However, at higher temperatures, they can either remain as spherical smooth surface particles, or turn into spherical bumpy particle or clubSP. The ability of E proteins to undergo some movements depends on the residues that are involved in the intra-dimer, inter-dimer and inter-raft interactions, and also the residues that are involved in the interactions between E protein ectodomain and stem helices of the E and M proteins.

Some highly neutralizing HMAb have been shown to recognize quaternary structure-dependent epitopes on the spherical smooth-surfaced particle; however, these antibodies may not be able to neutralize the bumpy-surfaced particles or clubSP. Conversely, antibodies that recognize partial or completely hidden epitope, such as fusion loop antibodies or DIII-binding antibodies, could bind well to bumpy-surfaced virus particles due to their looser E protein arrangement and the higher solvent accessibility of the epitopes. Bumpy morphology virus could thus likely elicit more cross-reactive fusion loop or DIII antibodies. ClubSP has a heterogenous way of displaying E protein epitopes—different parts of the clubSP have different levels of exposure of different epitopes. Altogether, this suggests that the morphological changes on the DENV particle may facilitate the virus to escape from the immune system. Therefore, it is important to understand all possible morphological structures of flavivirus in order to design effective therapeutics or vaccines.

## Figures and Tables

**Figure 1 viruses-13-01446-f001:**
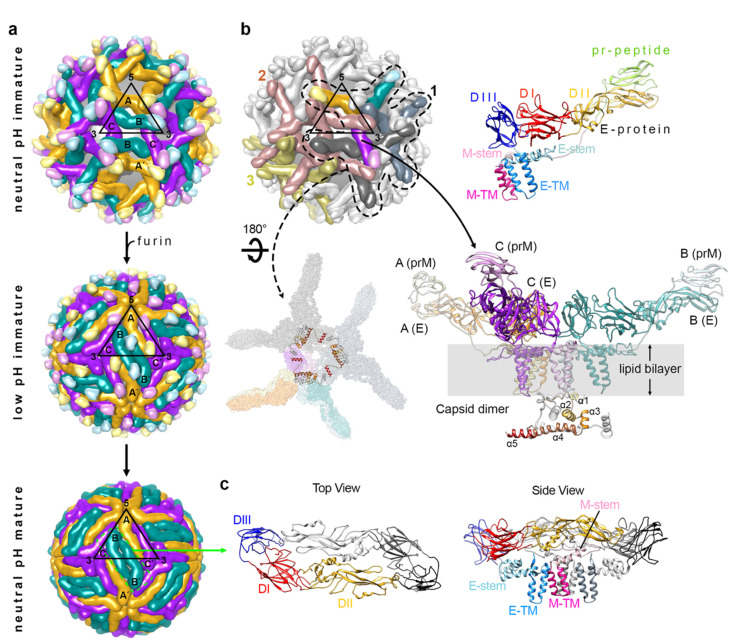
The organization of the structural proteins in DENV. (**a**) Different arrangement of the E protein or E-prM protein on neutral pH immature (top), low pH immature (center), and neutral pH mature (bottom) particles that are present in the maturation process of DENV. The black triangles indicate an icosahedral asymmetric unit (asu), with 5-, 3- and 2-fold vertices shown. Each asu contains 3 individual E proteins, which are indicated as molecules A, B and C. The E proteins in the neighboring asu are labeled A′, B′, and C′. They are each located near to 5-, 2- and 3-fold vertices and are coloured in gold, dark cyan, and purple, respectively. (**b**) E and prM proteins in immature DENV exist as a heterodimer, and three E-prM heterodimers form an inverted tripod structure. (Top, right) The E protein ectodomain consists of DI (red), DII (yellow) and DIII (dark blue), which is connected to membrane helices (blue) via three stem helices (cyan). PrM protein consist of pr-peptide (light green), a long loop that contains furin cleavage site, stem helices (pink) and then trans-membrane helices (dark pink). (Bottom, right) Three E-prM heterotrimers form an inverted tripod structure and a C protein dimer bound to the trans membrane helices region of the inverted tripod. The three E-prM heterodimers that form an inverted tripod are colored in gold, dark cyan, and purple. The helices α1–α5 of one of the C protein protomer in a dimer are colored in a gradient of brown shades, whereas the the other protomer is colored in gray. (Bottom, left) Three inverted tripod structure interact to each other to form the building block for the immature virus shell. The three C protein dimers further make a triangular network strengthening the building block structure. (Top, left) One building block interacts with another through the tips of their E–prM heterodimers forming the spiky-looking lattice of the immature virus. Three neighboring building blocks (numbered 1 to 3) are shown with colors whereas the others are in light gray. Building block 1 (indicated in dashed line) has one inverted tripod colored in gold-dark cyan-purple, whereas the other two inverted tripods are in two different shades of gray. Building blocks 2 and 3 are colored in brown and olive green, respectively. (**c**) An E protein dimer that presence on the mature DENV structure. One E protein protomer in a dimer is colored as in panel b (top, right), whereas the other E protein is colored in gray.

**Figure 2 viruses-13-01446-f002:**
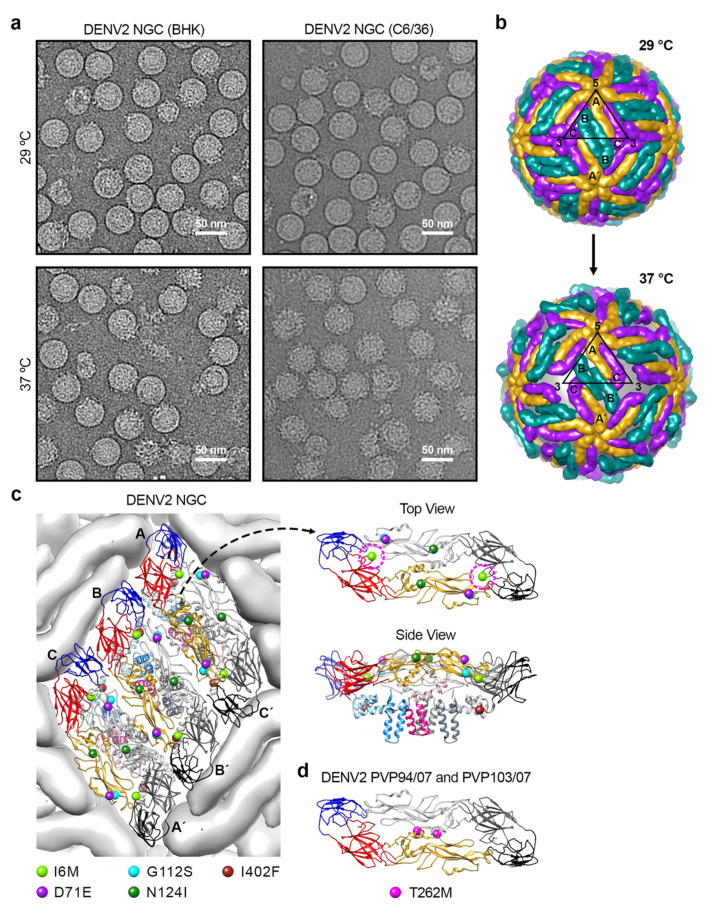
Structural changes in DENV2 strain NGC at 37 °C. (**a**) CryoEM micrographs of the lab-adapted DENV2 NGC strains with different passage histories in mammalian (BHK21) and mosquito (C6/36) cell lines, when incubated at 29 and 37 °C. (**b**) The E protein on DENV2 surface undergoes structural rearrangement upon incubation at 37 °C. The organization of E proteins on DENV2 at 29 °C (top) and 37 °C (bottom) is shown. At 37 °C, the E protein mols A-C′ particle remains as a dimer, but it undergoes translation and rotation, whereas the E protein molecules B and B′ move apart from each other, and thus are no longer dimers. Furthermore, the E protein molecule B/B′ is rotated upwards using the fusion loop as a pivot point, resulting in the protusion of DIII. The black triangle represents an icosahedral asymmetric unit with the corresponding 5-, 2-, and 3-fold vertices indicated. (**c**) DENV2 NGC (BHK21) showed differences in their E protein sequences at 5 positions when compared to DENV2-NGC (C6/36). (Left) The locations of these 5 residue differences are shown on an E protein raft on the virus surface. The residue mutations are shown as spheres. (Right) Top and side view of an E protein dimer showing the locations of the 5 residue differences. Top view of the E protein dimer only shows the ectodomain part, whereas the side view includes the TM regions. Residue 6 (I6M) is indicated by magenta dashed circle. (**d**) The location of one residue difference on E protein observed between DENV2 strain PVP94/07 and DENV2 strain PVP103/07. DENV2 strain PVP103/07 displayed morphological changes at 37 °C, whereas DENV2 strain PVP94/07 not. The E protein is colored as in Figure 1.

**Figure 3 viruses-13-01446-f003:**
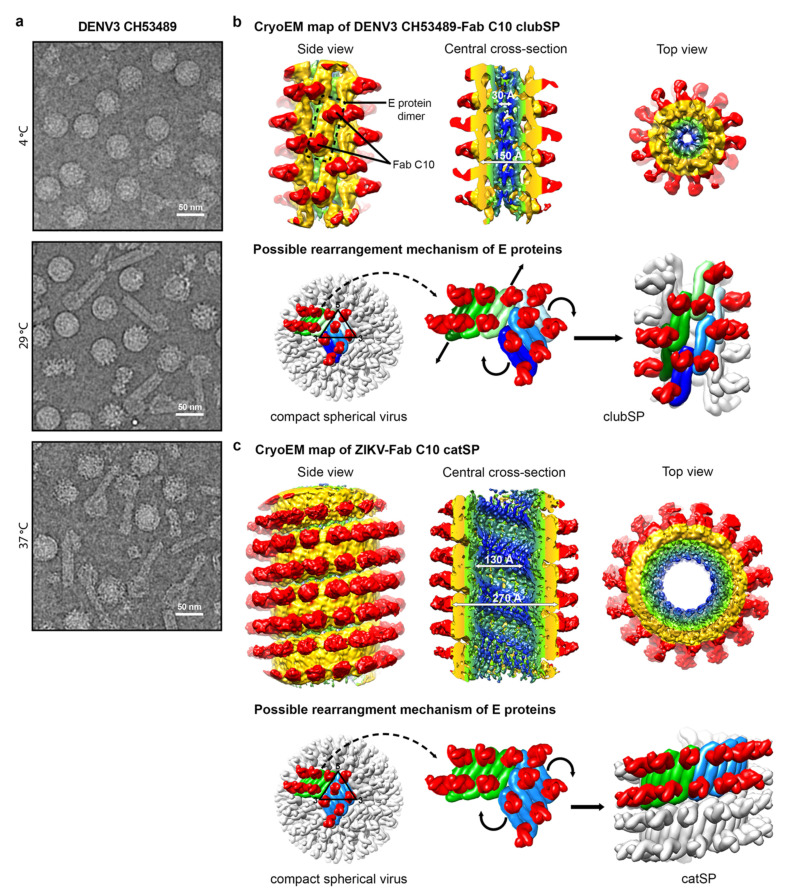
Helical structures of the tail of DENV3-Fab C10 clubSP and ZIKV-Fab C10 catSP. (**a**) CryoEM micrographs of the lab-adapted DENV3 CH53489 at 4, 29 and 37 °C. DENV3 at 4 °C contains mostly spherical particles, and when incubated at 29 and 37 °C, clubSP particles will form. In addition, at 37 °C, some of the smooth spherical-shaped particles also adopt a bumpy morphology. (**b**,**c**) (Top) The cryoEM map of the tail of DENV3-Fab C10 clubSP (**b**) and ZIKV-Fab C10 catSP (**c**). The side view (left), central cross-section (center), and top view of the reconstructed helical map are shown. For DENV3-Fab C10 clubSP, the map is colored according to the radii from the helical axis: blue (10–30 Å), green (31–50 Å), yellow (51–90 Å), and red (91–140 Å), whereas for ZIKV-Fab C10 catSP: blue (10–75 Å), green (76–105 Å), yellow (106–140 Å), and red (146–195 Å). These colors correspond to the inner leaflet of the lipid bilayer membrane, the transmembrane regions of the E and M proteins, the ectodomain of the E and M proteins, and Fab C10, respectively. (Bottom) The proposed mechanism of how the E proteins rearrange from the original smooth compact spherical virus structure to form the final helical structure. The DENV3 CH53489-Fab C10 clubSP structure (**b**) shows that Fab C10 does not lock the DENV3 raft structure. The Fab C10 only locks the E protein dimer and the dimers move away from each other. The orientation of each E protein dimer suggests that, at a higher temperature, the E protein rafts may rotate (indicated by curved black arrows in the center panel) relative to other rafts (as also observed in ZIKV-Fab C10 catSP, see below), and then the E protein dimers within a raft translate laterally away from each other (indicated by vertical arrows in center panel) to achieve the final helical structure arrangement. The lateral movement of E protein dimers indicates that inter-dimer interactions are also weak, in addition to the inter-raft interactions. Two rafts, each containing three dimers, are colored in different shades of green and blue. The ZIKV-Fab C10 catSP (**c**) structure indicates that all E proteins within a raft are locked together by Fab C10, as observed in the cryoEM structure of ZIKV-Fab C10 spherical particles [61]. The rafts in the spherical ZIKV particles (shown in green and blue, left panel), at increased temperature, rotate relative to each other (indicated by curved arrows in the center panel) in order to form the helical structure (right panel).

**Table 1 viruses-13-01446-t001:** DENV strains and their morphology at 37 or 40 °C.

Serotype	Strain	Morphology		Isolate	Reference
		37 °C	40 °C		
DENV1	PVP159	smooth		clinical	[50,51]
	WestPac74	~50% bumpy		lab adapted	[50]
DENV2	NGC (BHK21)	smooth		lab adapted	[58]
	NGC (C6/36)	bumpy		lab adapted	[46,50,51,58]
	S16803	bumpy		lab adapted	[46]
	S16681	bumpy		lab adapted	[57]
	PVP94/07	smooth	bumpy	clinical	[50,58,59]
	05K4155	smooth	bumpy	clinical	[58]
	PVP103/07	bumpy	bumpy	clinical	[58]
	SL56	smooth	bumpy	clinical	[58]
DENV3	863DK	smooth		clinical	[53]
	CH53489	club shape		lab adapted	[47]
DENV4	06K2270DK1	smooth	relatively smooth, aggregated	clinical	[51]

## Data Availability

All data is presented in references.

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
