# Peer review of "Morphological Diversity and Dynamics of Dengue Virus Affecting Antigenicity"

_viruses, 2021, doi:10.3390/v13081446_

Round 1

Reviewer 1 Report

Dengue virus (DENV) is a Flaviviridae member of important Human health concern. Producing a variety of symptoms since mild illness to acute flu-like illness or occasionally develops into a potentially lethal complication. DENV is transmitted to humans via the bite of an infected mosquito. There are four DENV serotypes (DENV-1, DENV-2, DENV-3 and DENV-4) that differ by ~ 25-40% at aminoacid sequence level.
The viral particle harbor the viral positive sense single stranded RNA genome complexed with the capsid protein (C) surrounded by a lipid bilayer anchored with membrane (M) and envelope (E) proteins. The last one is the major target for neutralizing antibodies.
In this manuscript, the authors reviewed the morphological changes of DENV serotypes found by different works including recent results from bibliography. This is an interesting review in wich they compare the 3D structural information (Cryo-EM structures) with biophysical results in order to better understand the dynamic of the structural changes of viral particle. This property is especially important in order to understand how the viruses escape from neutralizing antibodies in the human immune system. 

I only have some minor comments: 
- Figure 1b, top panel, pr-peptide must be substituted by pr-M.
- Figure 2b legend, line 226, the indicated temperature in text is 28 ° C but in the image is 29 ° C? Which is the right one? Also in Figure 2a you showed 29 vs 37 ° C is it ritght? Or is it 28 ° C? The temperature for the insect cells growth.
- lines 320, 525, Celsius symbol ºC must be replaced for “° C”
-Figure 3 legend, line 422, check the Celsius symbol that are as subscript.

Reviewer 2 Report

This manuscript by Fibriansah et al. reviewed all known morphological variants of DENV discovered thus far, their surface proteins dynamic and the key residues or interactions that play an important role in the structural changes, and how those morphological changes may alter the antigenicity characteristics of the virus particle, which in turn may affect the type of the elicited antibodies during an immune response. Although the manuscript has potential significance in dengue research, I found it lacks some important information to prove the conclusion. My comments are listed below: 

The abstract is poorly written. For example, why emphasized the temperature only in the abstract while the manuscript discusses pH, antibodies, etc.?

Lines 49-50: There is a lack of appropriate references to support this statement.

Lines 58-67: Lack of relevant references.

Lines 70-76: Lack of relevant references.

Lines 96-97 and lines 119-123: The statement states that “These immDENV particles are then transported through the trans-Golgi network where the pH environment gradually becomes acidic. The acidic pH triggers massive viral surface rearrangement – changing the morphology of the virus from spiky to smooth surface (Figure 1a). This structural arrangement involves the trimeric prM-E proteins disassociating from each other and then reassociating to form homodimers (Figure 1c). The resultant structure also causes the furin-cleavage site located on the stem between pr molecule and M protein to become more exposed, allowing cleavage to occur.” And then lines 128-130 state, “The maturation process of DENV is inefficient, as we often observe the presence of mature, partially mature and highly immature virus particles in a mature virus preparation [27,28].” Why is the maturation process of DENV inefficient? Is it because the pH level at the trans-Golgi is not low enough for the rearrangement to happen? Also, according to the statement, the acidic pH triggers massive morphological rearrangement of the viral surface proteins to change from spiky to smooth. This change allows the cleavage of the pr from the M protein to occur. Does cleavage of the pr portion help stabilize the smooth appearance of the virus after exiting the cell? Why is this cleavage of the prM important for the maturation process of the dengue?

The statement strongly implies that the smooth surface of the virus particle is dependent on the low pH at the trans-Golgi apparatus.  What is the difference between mature and smooth virus particles? Is it means that the prM proteins were all cleaved and then rearranged to have bumpy surfaced particles when exposed to 37°C or 40°C?

Lines 123-126: The quoted reference does not support the statement. Dengue virus particles have never been demonstrated to be transported via exosomes.  

Lines 130-133: The statement says, “As even the highly immDENV has been shown to be infectious to DC-SIGN bearing cells [29] and also to Fc-gamma receptor-positive cells when immDENV is complexed with some anti-E and anti-prM antibodies [30,31], therefore all viruses with different degrees of maturation is considered infectious.” This is a very misleading statement that may confuse the readers. However, the statement may be true if an individual has been exposed to dengue before infection by the immDENV of the possibility of the presence of the anti-prM or anti-E antibodies. Also, does it matter whether the prM and E antibodies are of different serotypes?

Lines 133-136: Rewrite to clearly state that it is a hypothetical statement.

Lines 148-156: Rewrite this paragraph. This paragraph is like comparing an orange to an apple.

Lines 160-162: The statement states that “Other strains produce virus with the mature compact DENV at mosquito physiological temperature (28°C), but changes to a bumpy surface structure when incubated at higher temperature (37°C or 40°C) [32,37]. Does this statement mean that the prM proteins are all cleaved in mosquitoes, and the particles remain smooth in 28oC but change to bumpy when exposed to higher ToC? How about the virus particles collected from a different cell line?

Lines 70-126: You need to reference all the relevant published data.

Lines 297-316: These two paragraphs are poorly written and are both very confusing. Need major revision.

Lines 451-455: Lack of relevant references.

Lines 245-248: The statement states that “Examination of four naturally circulated DENV2 strains (05K4155, PVP94/07, 245 PVP103/07 and SL56) showed that at 37oC, the surface of three of these DENV2 strains 246 (05K4155, PVP94/07 and SL56) remained smooth, whereas strain PVP103/07 turned into 247 bumpy morphology [46]. Based on the conclusion of the manuscript, “Are you saying that the PVP103/07 was more virulent than PVP94/07 because of its bumpy morphology? If so, is there clinical data to support this argument?

Lines 269-280: Most lab-adapted viruses, with a long passage history in C6/36 mosquito cell lines at 28oC, have bumpy surface morphology, while the clinical DENV isolates which circulate between human and mosquito hosts are largely smooth surface particles. This suggests that the laboratory passage conditions do not have the selection pressure for smooth surface particles. The observation that many different E protein mutations [46] can result in bumpy surface particles at 37oC suggests that during passaging virus for numerous times in mosquito cell lines at 28oC, some mutations have been accumulated that lead to the majority of laboratory strains to acquire the bumpy surface morphology. In contrast, it was observed that 3 out of 4 clinical DENV isolates are smooth-surfaced, suggesting there is selection pressure towards the smooth surface particles. However, one strain is bumpy surfaced, suggesting that the naturally circulating virus could change morphology if there needed, i.e., to evade the immune system.

This is a very confusing argument because you are saying that most labs adapted viruses have bumpy surface morphology (which is beneficial for the virus), but they do not have the selection pressure for smooth surface particles (which is beneficial for the host), and this lab adapted virus has accumulated mutations. And then you argued that one out of four of the clinical DENV isolates is bumpy to evade the immune system. Which of the two conditions that drive the DENV better to evade immune systems, the laboratory passage conditions or inside the host? Please explain.

Lines 269-283: Lack of relevant references.

Lines 297-309: The observations described have no references. Are these observations new????

Lines 300-304: The statement is so confusing. For example, it states that “However, it can observe the overall average changes of the deuterium exchange of E proteins on the virus surface. The DENV2 strain NGC, which is observed to have bumpy surface morphology under cryoEM, seems to have more motions detected by HDXMS in E, M, and C proteins than that of DENV1 strain PVP159 at both 28°C and 37°C”.   Why include C when the focus of the manuscript is on the E and M? Why are you comparing the bumpy surface DENV2 strain NGC (isolated from the mosquito cells) to DENV1 strain PVP159 but not the smooth, compact DENV2 strain NGC (isolated from mammalian cells)?

Round 2

Reviewer 2 Report

Minor spelling and grammar check is required.